



# Dynamics of ENSO-driven stratosphere-to-troposphere transport of ozone over North America

John R. Albers[1,2], Amy H. Butler[3], Andrew O. Langford[3], Dillon Elsbury[1,3], Melissa L. Breeden[1,2]

[1]Cooperative Institute for Research in the Environmental Sciences, University of Colorado Boulder, Boulder, 80305, USA
[2] NOAA Physical Sciences Laboratory, Boulder, 80305, USA
[3] NOAA Chemical Sciences Laboratory, Boulder, 80305, USA

*Correspondence to*: John R. Albers (john.albers@noaa.gov)

**Abstract.** The El Niño-Southern Oscillation (ENSO) is known to modulate the strength and frequency of stratosphere-to-troposphere transport (STT) of ozone over the Pacific-North American region during late winter to early summer. Dynamical
processes that have been proposed to account for this variability include: variations in the amount of ozone in the lowermost stratosphere that is available for STT, and tropospheric circulation-related variations in the frequency and geographic distribution of individual STT events.

Here we use a large ensemble of Whole Atmosphere Community Climate Model (WACCM) simulations (forced by sea-surface temperature (SST) boundary conditions consistent with each phase of ENSO) to show that variability in lower
stratospheric ozone and shifts in the Pacific tropospheric jet constructively contribute to the amount of STT of ozone in the North American region during both ENSO phases. In terms of stratospheric variability, ENSO drives ozone anomalies resembling the Pacific-North American teleconnection pattern that span much of the lower stratosphere below 50 hPa. These ozone anomalies, which dominate over ENSO-driven changes in the Brewer-Dobson circulation (including changes due to both the stratospheric residual circulation and quasi-isentropic mixing), strongly modulate the amount of ozone available for
STT transport. As a result, during late winter (February-March), the stratospheric ozone response to the teleconnections constructively reinforces anomalous ENSO-jet-driven STT of ozone. However, as ENSO forcing weakens as spring progresses into summer (April-June), the direct effects of the ENSO-jet-driven STT transport weaken. Nevertheless, the residual impacts of the teleconnections on the amount of ozone in the lower stratosphere persist, and these anomalies in turn continue to cause anomalous STT of ozone. These results should prove helpful for interpreting the utility of ENSO as a
subseasonal predictor of both free-tropospheric ozone and the probability of stratospheric ozone intrusion events that may cause exceedances in surface air quality standards.

## 1 Introduction

Ozone transported from the stratosphere contributes to the North American background (NAB) ozone concentration in the free troposphere (Fiore et al., 2014; Cooper et al., 2015; Young et al., 2018) and to surface ozone exceedance events that
affect human health (Fiore et al., 2014; Cooper et al., 2015; Young et al., 2018; Zhang et al. 2020; Langford et al. 2022).





Unfortunately, estimating the stratospheric contribution to surface exceedances and the NAB is quite complex because atmospheric internal variability and low-frequency climate modes (e.g., the El Niño-Southern Oscillation and the quasi-biennial oscillation) combine to drive significant subseasonal-to-seasonal variations in stratosphere-to-troposphere transport.

On subseasonal-to-seasonal timescales, variability in stratosphere-to-troposphere transport (STT) primarily modulated via two dynamical processes (Albers et al. 2018 and references therein): variations in the amount of ozone in the lowermost stratosphere available for STT, and tropospheric circulation-related variations in the frequency, depth, and geographic distribution of individual STT events (e.g., Breeden et al. 2021). The El Niño-Southern Oscillation (ENSO) is thought to modulate both processes, but unfortunately, prior research yields somewhat conflicting results. For example, on hemispheric spatial scales, Neu et al. (2014) used Tropospheric Emission Spectrometer (TES) and Microwave Limb Sounder (MLS) data (2005-2010) and suggested that the warm phase of ENSO accelerates the Brewer-Dobson circulation (BDC), which leads to more ozone in the lowermost midlatitude stratosphere, and subsequently causes an increase in STT of ozone into the midlatitude troposphere (see also, García-Herrera et al. 2006, Calvo et al. 2010, and Simpson et al. 2011). This view is supported by Zeng and Pyle (2005) who found a positive correlation between ENSO and global STT for 1990-2001. On the other hand, Hsu and Prather (2009) find a weak correlation between hemispheric or global STT and ENSO during a slightly later period (2001-2005). If more localized spatial scales are considered, some conflicting results remain. For example, Lin et al. (2015) suggested that La Niña (1990-2012) shifts the Pacific storm track northward and increases its variability, leading to more frequent deep stratospheric intrusions transporting stratospheric ozone into the lower troposphere over western North America. However, Langford (1999) found that El Niño (1993-1998) extends the subtropical jet eastward, driving transverse circulations at the nose of the jet that can also increase ozone transport into the middle and upper troposphere over western North America.

It is difficult to discern whether the aforementioned results appear to conflict because of the relatively short data records used, or whether both phases of ENSO can potentially increase STT of ozone, but with some sensitivity to the specific geographic region. Here we address both possibilities by using a large ensemble of Whole Atmosphere Community Climate Model (WACCM) simulations to quantify how ENSO modulates subseasonal variations in STT of ozone over the Pacific-North American region. The WACCM simulations reveal that ENSO-driven changes in STT are highly dependent on the time of year and geography, with stratospheric and tropospheric processes combining to increase STT in one region while decreasing STT in a second region during the same ENSO phase. Moreover, the large number of simulated ENSO years, forced by the same sea surface temperature patterns in order to reduce influence of ENSO diversity, allows quantification of the relative importance of ENSO-induced changes in the midlatitude (Lin et al. 2015) and subtropical (Langford 1999) jets for modulating STT of ozone. And in contrast to some previous studies suggesting that the BDC is the mediating link between ENSO and stratospheric changes in STT, our results suggest that it is ozone teleconnections that are most important, a finding which is consistent with Zhang et al. 2015 and Olsen et al. 2016.

Ozone teleconnections, first recognized by Reed (1950) (see also Schoeberl and Krueger 1983, Mote et al. 1991, Stephenson and Royer 1995, Wang et al. 2011), result from vertical motion and horizontal advection induced by planetary





wave geopotential height perturbations that are associated with opposite signed ozone perturbations (i.e., a positive geopotential height anomaly is associated with a negative ozone anomaly). In the WACCM simulations, ENSO drives ozone anomalies resembling the Pacific-North American teleconnection pattern that extend from the lowermost stratosphere to at least 50 hPa in height. Here we provide a detailed analysis using a stratospheric ozone tracer ($O_3S$ from Tilmes et al. 2016), to demonstrate how the jet shifts and ozone teleconnections patterns caused by ENSO constructively contribute to drive

changes in STT of ozone over North America

Section 2 outlines the WACCM simulations used, Section 3 details how stratospheric and tropospheric processes constructively reinforce anomalous STT during different times of the seasonal cycle and for different portions of Pacific-North American region. A discussion of the implications of our results for subseasonal prediction of STT of ozone is contained in Section 4.

**2 Climate model simulations**

**2.1 WACCM simulations**

Simulations were created using the National Center for Atmospheric Research Whole Atmosphere Community Climate Model (WACCM) version 4 (Mills et al. 2017). WACCM has fully interactive chemistry in the middle atmosphere, which

includes a stratospheric ozone tracer ($O_3S$) that evolves via full chemistry in the stratosphere and then decays at a tropospheric chemistry rate once it crosses the tropopause (the version of WACCM we use here includes ozone removal via tropospheric dry deposition). The $O_3S$ tracer should be interpreted to represent an upper-bound of the stratospheric contribution to a stratosphere-to-troposphere ozone fold, in large part because it is missing some tropospheric chemistry that would likely decrease its tropospheric chemical lifetime (Emmons et al. 2003). WACCM has a limited representation of

tropospheric chemistry, but simulates background tropospheric interannual ozone variability quite well (Hess et al. 2015). The model extends to ~140 km with 70 vertical levels, and a horizontal resolution of 1.9 degrees latitude by 2.5 degrees longitude.

We conducted two sets of "time-slice" simulations, one set each for El Niño and La Niña conditions. Each simulation is 60 years long, with the first 10 years of each simulation used as model 'spin-up' time and subsequently discarded. The

simulations are forced via SST composites for each ENSO phase, which were created by averaging over all El Niño and La Niña events (defined as Niño 3.4>1 standard deviation from the March long-term mean) using Hadley Centre Global Sea Ice and Sea Surface Temperature data (HadISST2, years 1950-2008; see https://climatedataguide.ucar.edu/climate-data/nino-sst-indices-nino-12-3-34-4-oni-and-tni for a description of the Niño 3.4 index).  SST anomalies evolve over a two-year cycle, growing from zero SST anomaly in January of year one, to a peak anomaly in January of year two, and then back to zero

anomaly by the end of December of year 2. The SSTs anomalies are tapered in space via a half-cosine weighting function so that there is no SST anomaly poleward of +/- 25° latitude (see Supplementary Fig. S1 for complete cycle). The ENSO SST composite was constructed to reproduce the largest March SST anomalies in the observational record, thus allowing us to




establish an upper bound on the potential effects of ENSO-related control of spring season STT. However, even when constructed in this way, the SST anomalies still peak in January of the second year of the two-year ENSO cycle. All

composites are computed for year two of the two-year ENSO cycle, which corresponds to the mature and decaying portion of the ENSO cycle. In all cases, radiative forcings, tropospheric emission precursors, and volcanic sulphates were set to preindustrial (year 1850) levels, to avoid major influence from greenhouse gases or ozone-depleting substances. The simulations contain no tropical quasi-biennial oscillation, and for solar and geomagnetic parameters, a solar cycle average was used. WACCM anomalies are created by subtracting a climatology that is created from the average over all ensembles

(El Niño and La Niña simulations, i.e., 100-years of data).

Eliassen-Palm flux vectors and the stratospheric residual circulation ($\bar{v}^*, \bar{w}^*$) are calculated on daily timescales using the Transformed Eulerian Mean formulation in spherical coordinates as defined in Andrews et al. (1987). Eddy kinetic energy ($1/2[u'^2 + v'^2]$) is calculated from daily data that is band-passed filtered. For eddy kinetic energy (EKE) near the tropopause, a 2-10 day filter is applied to highlight synoptic timescale variability, while for EKE in the upper stratosphere, a

slightly broader 2-120 day filter is applied to capture synoptic and more slowly evolving planetary scale eddy variability (e.g., Albers et al. 2016).

In the results, most figures are shown for both ENSO phases (or the difference of El Niño minus La Niña) or are included in the Supplement. In all cases, the La Niña anomalies are essentially identical to the El Niño anomalies, but opposite in sign, which is due to the symmetry in the prescribed Niño 3.4-based SST forcing for both ENSO phases.

**3 Results**

The climatological seasonal cycle of Northern Hemisphere (and North American) extratropical and high latitude STT of ozone in the WACCM time slice simulations is consistent with observations (e.g., Lefohn et al. 2001, Albers et al. 2018), with notable stratospheric ozone ($O_3S$) present in the lower troposphere beginning in December, $O_3S$ peaking in March and April, and then largely decreasing so that negligible $O_3S$ anomalies are present by mid-summer (Supplement Fig. S2).

Interannual variability in high latitude total column ozone can largely be accounted for by variability in the wave-driven BDC, with stratospheric chemistry accounting for less than 20% of the interannual variance (Fusco and Salby 1999, Salby and Callaghan 2002, Weber et al. 2011). The vertical distribution of ozone within the column is controlled by a combination of the BDC (including both advective and eddy transport) in the stratosphere during winter and stratospheric intrusions along the tropopause during spring and early summer. Within the stratosphere, extratropical stratospheric planetary wave driving

peaks in Northern Hemisphere winter (Charney & Drazin, 1961; Randel et al., 2002), which increases ozone transport along the deep branch of the BDC leading to a seasonal build-up of ozone in the high latitude lowermost stratosphere (Konopka et al. 2015, Ploeger and Birner, 2016, Ray et al. 1999, Bönisch et al. 2009, Butchart 2014, Hegglin and Shepherd 2007). As spring proceeds into early summer, eddy activity (and hence stratospheric intrusion frequency) along the extratropical tropopause increases in vigor (Breeden et al. 2021 and references therein), which leads to a "flushing" of ozone from the



stratosphere into the troposphere that accounts for the seasonal peak in STT of ozone over North America (James et al. 2003, Lefohn et al. 2001, Škerlak et al. 2014, Terao et al. 2008, Albers et al. 2018).

## 3.1 Stratospheric circulation

During late winter, El Niño accelerates the BDC along both the shallow and deep branches (Fig. 1a; the qualitatively, but opposite signed La Niña anomalies are shown Fig. 1c). Along the shallow branch (below roughly 70 hPa), there is strong upwelling between $0°$-$10°$ N and strong downwelling between $10°$-$30°$ N, which is driven by a combination of planetary and gravity waves in the subtropics as documented in previous studies (Garcia-Herrera et al. 2006, Calvo et al. 2010, Simpson et al. 2011, Diallo et al. 2019). Along the deep branch, there is enhanced poleward flow above 10 hPa (not shown) and

enhanced downwelling poleward of $30°$ N (primarily above 40 hPa), which is consistent with the observed relationship between the Niño 3.4 index and the BDC (e.g., Rao et al. 2019 and references therein). By mid-spring the enhanced residual circulation along the shallow branch of the BDC persists (Figs. 1b and 1d), though the vertical extent is reduced, while the ENSO-induced changes along the deep branch of the BDC become negligible.

    The accelerated residual circulation along the deep branch of the BDC reflects, in part, an increase in planetary wave

driving in the extratropical upper stratosphere (not shown), which is dominated by planetary wavenumber one (Fig. 2a, see also Li and Lau 2013 and Rao et al. 2019 for similar results). In addition to accelerating the residual circulation, the anomalous planetary waves are also associated with enhanced wave breaking (McIntyre and Palmer 1983) as evidenced by anomalously large EKE spanning higher latitudes from the North Atlantic eastward to the Kamchatka Peninsula (Fig. 2b). Because the latitudinal gradient of $O_3S$ is negative ($O_3S$ decreases poleward, Fig. 3 right columns), the enhanced residual

circulation and planetary wave breaking (and associated quasi-isentropic mixing) increases $O_3S$ at high latitudes and decreases $O_3S$ in the extratropics (Fig. 2c). At the beginning of February, this $O_3S$ pattern is seen as the north-south dipole structure between 5-10 hPa in Fig. 3a. As spring proceeds, that $O_3S$ anomaly pattern is advected poleward and downward by the enhanced residual circulation so that the positive anomaly that is initially located between $80°$-$90°$ N and 5-10 hPa during February (Figs. 3a-b) is advected downward to 20-30 hPa by April (Figs. 3e-f), and to 40-50 hPa by May and June (Figs. 3g-

j). However, these anomalies never reach the lowermost stratosphere, and thus appear to be of negligible importance to spring season STT of $O_3S$. This raises the question, if it is not the deep branch of the BDC (i.e., the combined effects of quasi-isentropic mixing and the residual circulation) that govern the ENSO-induced $O_3S$ anomalies in the lowermost stratosphere, then what is responsible for the broad increase in extratropical $O_3S$ anomalies in the lowermost stratosphere from winter to spring during El Nino (Fig. 3, left column)?

One possible explanation is hinted at in NASA Microwave Limb Sounder and GEOS-chemistry climate model simulations shown in Oman et al. 2013 and Olsen et al. 2016, where total column ozone was regressed onto the Niño 3.4 index to reveal a faint pattern of anomalies extending outwards from the tropics towards North America (see their Figs. 6 and 4, respectively). Similarly, Zhang et al. (2015) use National Institute of Water and Atmospheric Research column ozone





(an observational product that assimilates satellite measurements) to show a similar pattern during January to March, though

their column ozone patterns are a bit more difficult to interpret. Nevertheless, Zhang et al.'s suggestion that lower stratospheric ozone anomalies arise due to teleconnections forced by ENSO convection is supported by our WACCM simulations. However, the WACCM results considered here suggest that the impact of the teleconnections extends far into the interior of the stratosphere and are thus not due solely to anomalies in tropopause height as suggested by Zhang et al. In particular, the $O_3S$ dipole between 20-200 hPa and poleward of 40° N (Figs. 3a-d) is almost completely explained by

vertically deep teleconnections, where geopotential height and $O_3S$ are almost perfectly anticorrelated (cf. Figs. 4a and 4b and Figs. 4c and 4d, both on the 200 hPa surface). Similar geopotential height-$O_3S$ patterns are observed in our WACCM simulations all the way up to 20 hPa (not shown), where the geopotential height and $O_3S$ patterns transition to the wavenumber one structure shown in Fig. 2.

The anticorrelated geopotential height-$O_3S$ patterns (Fig. 4) responsible for high latitude $O_3S$ dipole between 30-200 hPa

(Figs. 3a-d) are consistent with the vertical motion and horizontal advection explanation first described and modeled by Reed (1950) using single column vertical profiles of observed ozone and atmospheric circulation. The underlying dynamics of the synoptic scale wave-ozone relationship suggested by Reed were later confirmed by Schoeberl and Krueger (1983) using Nimbus 7 Total Ozone Mapping Spectrometer ozone data and First Global GARP Experiment (FGGE) temperature (geopotential height) data (see also Salby 1982). Schoeberl and Krueger conclude with several findings that are relevant to

the current WACCM results, namely: (1) for medium-scale waves, geopotential height and ozone anomalies should be anticorrelated; (2) vertical motion and horizontal advection are of equal importance to generating the ozone anomalies; and (3) evanescent waves in the lower stratosphere should produce the maximum ozone signal because they will have minimal phase tilt with height and thus vertical motion and horizontal advection will cause additive anomalies in ozone. The signature of the medium-scale geopotential height waves shown in Figs. 4b and 4d are part of wave structures that exhibit

very little phase tilt with height (Fig. 5), which is consistent with waves that are largely evanescent (Charney and Drazin 1961). Thus, the anticorrelated (180° out-of-phase) geopotential height-$O_3S$ patterns shown in Fig. 4 are consistent with the findings of Schoeberl and Kreuger, suggesting that the $O_3S$ anomalies are caused by vertical motion and horizontal advection associated with medium-scale evanescent waves (i.e., planetary wavenumber >2). This is in contrast to what would be expected from longer scale vertically propagating planetary waves (planetary wavenumber <3), which are typically

associated with significant poleward and downward eddy-ozone flux transport in the upper stratospheric photochemical transition region where ozone and geopotential height are 180° out-of-phase, but cause minimal transport in the lower stratosphere where ozone and geopotential height tend to be close to in-phase (e.g., Fig. 8 of Hartmann and Garcia 1979; see also Garcia and Hartmann 1980, Gille et al. 1980, Hartmann 1981, Albers and Nathan 2012).

The geopotential height teleconnections, and associated $O_3S$ anomalies, peak in February and March (cf. Figs. 3a-d and

Fig. 4 for February-March) and quickly decay thereafter (cf. Figs. 3e-h and Fig. 6 for April-May). As the teleconnections dissipate, the large positive $O_3S$ anomaly that was once located over the North Pacific (Fig. 4a) is mixed northward, leading to weak positive $O_3S$ anomalies over most of the Northern Hemisphere poleward of 50° N (Figs. 3e-j and 6a). This poleward





mixing reflects the seasonal cycle of the stratospheric polar vortex and the build-up and breakdown of the polar transport barrier, which proceeds as follows (e.g., Manney et al. 1994). During midwinter, the polar night jet is typically located in the
mid- to upper stratosphere, which, because of the associated strong vortex edge potential vorticity gradient, creates a barrier to latitudinal transport. As spring onsets, the upper portions of the vortex weaken and the polar night jet descends and establishes a transport barrier in the lowermost stratosphere. However, once the stratospheric final warming occurs – typically sometime in March or April (Butler and Domeisen 2021) – the transport barrier is erased and mixing between polar and midlatitude air rapidly ensues (Manney et al. 1994, Salby and Callaghan 2007a,b), hence the anomalies shown in Figs.
3e-j and 6a. In the next section, we outline how the ENSO-induced O$_3$S anomalies just described constructively reinforce ENSO-induced changes in synoptic wave activity (and hence stratospheric intrusions) that modulate STT.

**3.2 Upper tropospheric-lower stratospheric circulation and STT**

Lin et al. (2015) suggested that deep STT of ozone over western North America should increase when La Niña (measured by the Niño 3.4 index) perturbs the polar front jet northward and invigorates it so that the frequency of deep tropopause folds increases. Consistent with this hypothesis, Breeden et al. (2021) find that in reanalysis, EKE and deep mass transport (though not necessarily ozone transport) over the western US increase during time periods when the Niño 3.4 index is at least moderately negative (<0.5° C). However, Breeden et al. further show that most of the ENSO jet-related changes occur in late
winter and early spring before the Pacific jet structure transitions from its winter to summer-like state (i.e., the jet transitions from being strong and zonally contiguous to being weak, with a discontinuity over the Pacific basin, e.g., Breeden et al. Fig 2). As a result, tropopause fold depth and frequency are increased primarily in February to mid-April during La Niña, and there are only smaller jet related changes in transport thereafter.

In agreement with the observed relationship between ENSO and EKE (e.g., Breeden et al. 2021, see their Supplemental
Fig. S6), the WACCM El Niño simulations show decreased EKE over the North Pacific-western US and increased EKE over Baja California-southern US during February and March (Fig. 7a), and vice versa for La Niña. The lower EKE to the north and higher to the south reflects the tendency for the time mean Pacific jet to shift southwards during El Niño and northward during La Niña (Shapiro et al 2001). Here, as in Shapiro et al. (2001), the time mean tends to convolve aspects of the subtropical and polar front jets (see also, Koch et al. 2006); however, what is important here, is that anomalies in EKE are
well-correlated with anomalous STT (Shapiro 1980, Langford 1999). Thus, comparing regions with anomalously high or low EKE (Fig. 7a) and high or low O$_3$S in the lowermost stratosphere (Figs. 4a and 4c) allows us to assess whether changes in tropopause fold frequency and/or transverse circulations near the nose of the jet, and stratospheric changes in the amount ozone available for downward transport, respectively, act constructively or in opposition to generate the observed anomalies of O$_3$S in the lowermost troposphere (Fig. 8 shows the case for El Niño; see Supplement Fig. S3 for the complementary La
Niña composites).





During February and March, there is anomalously high STT of $O_3S$ to the lower troposphere over the North Pacific (Figs. 8a and 8c). Because EKE is reduced over this region (Fig. 7a), which in isolation should correspond to a reduction in tropopause fold frequency, the anomalously high stratospheric $O_3S$ availability (Fig. 4a) must be the controlling factor governing the enhanced $O_3S$ transport into this region. In contrast, over the western US, reduced EKE and anomalously low

$O_3S$ operate constructively to reduce STT of $O_3S$. Similarly, though opposite in sign, the El Niño-induced increases in EKE and lower stratospheric $O_3S$ constructively contribute to enhanced deep STT of $O_3S$ over Baja and the southeastern US. By April-May, the lower stratospheric $O_3S$ anomalies and EKE and over the western US have weakened considerably (Figs. 6a and 7b, respectively), which is reflected in anomalous $O_3S$ only reaching the mid-troposphere (Figs. 8e-h). However, enhanced EKE and anomalously high lower stratospheric $O_3S$ continue to contribute to robust deep STT of $O_3S$ over Baja

and the southeastern US. By June, the peak in deep STT of $O_3S$ has receded westward and shifted northward so that the maximum transport is located over Baja and the southwestern US (Figs. 8i,j). These results make clear that the sign of anomalous STT of $O_3S$ due to ENSO is highly dependent on the region and time of year. For example, for the western US between 30°- 45° N, February-March El Niño conditions suppress the transport of stratospheric $O_3S$, but as spring progresses into summer (April-June), El Niño instead enhances $O_3S$ transport.

**4 Conclusions**

STT of ozone is modified by both stratospheric ozone variability and tropopause level jet dynamics (e.g., Lin et al. 2015, Albers et al. 2018, Langford et al. 2022 and references therein). Understanding the relative importance of these two processes is critical for both subseasonal prediction (Lin et al. 2015, Albers et al. 2018, 2021), as well assessing the interannual variability of tropospheric ozone concentrations relevant to the NAB (Fiore et al. 2003, 2014).

Our results suggest that there is no conflict between the results of Lin et al. (2015) versus Langford (1999) who hypothesize that La Niña versus El Niño, respectively, enhance western US STT of ozone during spring. Indeed, the two hypotheses can be reconciled by carefully accounting for the way that tropical-to-extratropical teleconnections modulate both lower stratospheric ozone availability and jet-related transport variability, which are both strongly dependent on the geographic region and month of the year under consideration. For example, during February and March, our results confirm

the hypothesis of Lin et al. (2015) suggesting that when La Niña conditions are present (as measured by the Niño 3.4 index), there will be enhanced STT over the western US (Supplemental Fig. S3). However, in contrast to Lin et al., our results here suggest that both tropospheric jet dynamics and stratospheric ozone availability contribute to the anomalous transport. At the same time, our results also confirm the hypothesis of Langford (1999), as we find that El Niño conditions lead to enhanced STT of ozone over Baja and the southern US during February and March (Figs. 7a-d), and enhanced ozone transport over the

western US during May and June (Figs. 7g-j). Again, our results suggest that in the later spring period, both tropospheric jet processes and stratospheric ozone availability constructively reinforce each other.

In addition, the relatively large ensemble of WACCM time slice simulations cleanly demonstrate how tropical-extratropical teleconnections cause zonally asymmetric anomalies in lower stratospheric ozone, which subsequently strongly





modulate STT of ozone. In particular, as suggested by Reed (1950) and Schoeberl and Krueger (1983), medium-scale
evanescent waves, which are largely barotropic, drive vertical motions and horizontal advection that lead to ozone anomaly
patterns that are nearly perfectly anticorrelated with geopotential height teleconnection patterns (Fig. 4). As a result, our
findings here confirm that ENSO-driven ozone teleconnections are the dominate mechanism controlling lower stratospheric
ozone changes relative to ENSO-driven changes in the Brewer-Dobson circulation, in agreement with the findings of Zhang
et al. (2013).

The sensitivity of ozone transport to the trajectory of tropical-extratropical teleconnections highlights a problematic
aspect of attempting to use ENSO indices (e.g., Niño 3.4) to make subseasonal-to-seasonal predictions of STT of ozone. For
example, a small northward shift in the location of the wave train depicted in Fig. 4 may change the sign of the western US
ozone transport anomaly altogether (i.e., the positive ozone anomaly over Baja could displace the negative ozone anomaly of
the western US in Figs. 7a-d). Indeed, even during times when the Niño 3.4 index is strongly loaded, ENSO diversity
(Capotondi et al. 2015) can yield distinctly different teleconnection patterns (e.g., Garfinkel et al. 2013). Moreover, when
ENSO teleconnections are convolved with internal variability, the resulting anomaly patterns over North America can vary
significantly (e.g., Deser et al. 2017, 2018), which further complicates using ENSO to predict STT of ozone. Thus, instead of
using a Niño-based index as a predictor, a perhaps more reliably method is to use a measure of signal or signal-to-noise ratio
to identify time periods when teleconnections and STT may be more predictable (e.g., Albers and Newman 2019, Albers et
al. 2021).

**Code and Data Availability**

The code used to perform this analysis can be accessed by personal communication with the corresponding author. The
WACCM simulation data used to create figures can be accessed here:
https://csl.noaa.gov/groups/csl8/modeldata/data/Albers_etal_2022/ .

**Author Contributions**

JRA wrote climate model analysis code, created the figures, and wrote the manuscript. AHB conducted the climate model
experiments. AHB, DE, MLB, and AOL provided comments and edited the manuscript.

**Competing Interests**

The authors declare no conflict of interest.

**Financial Support**

John R. Albers and Dillon Elsbury were funded in part by National Science Foundation grant #1756958.



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



**Figure 1:** Transformed Eulerian mean (TEM) residual circulation (red arrows) and TEM residual vertical velocity ($\overline{w}^*$, filled color contours) for: (a) February-March and (b) April-May of the El Niño time slice simulations; and (c) February-March and (d) April-May of the La Niña time slice simulations. Units for the residual circulation are in mm s$^{-1}$.





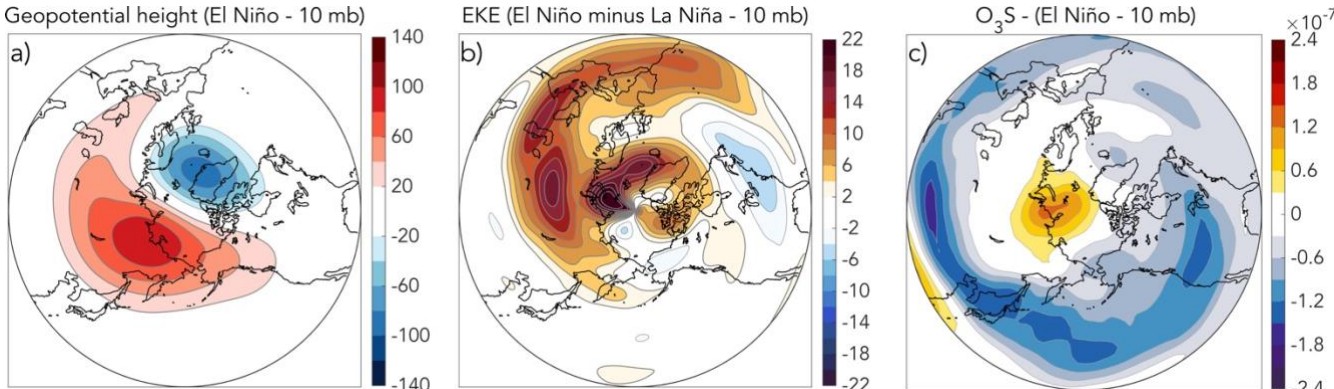

**Figure 2:** February-March 10 hPa composites of (a) geopotential height (units: gpm) for El Niño, (b) eddy kinetic energy (2 to 120-day filtered with units: $m^2 s^{-2}$) for El Niño minus La Niña, and stratospheric $O_3S$ (units: mass mixing ratio) for El Niño.





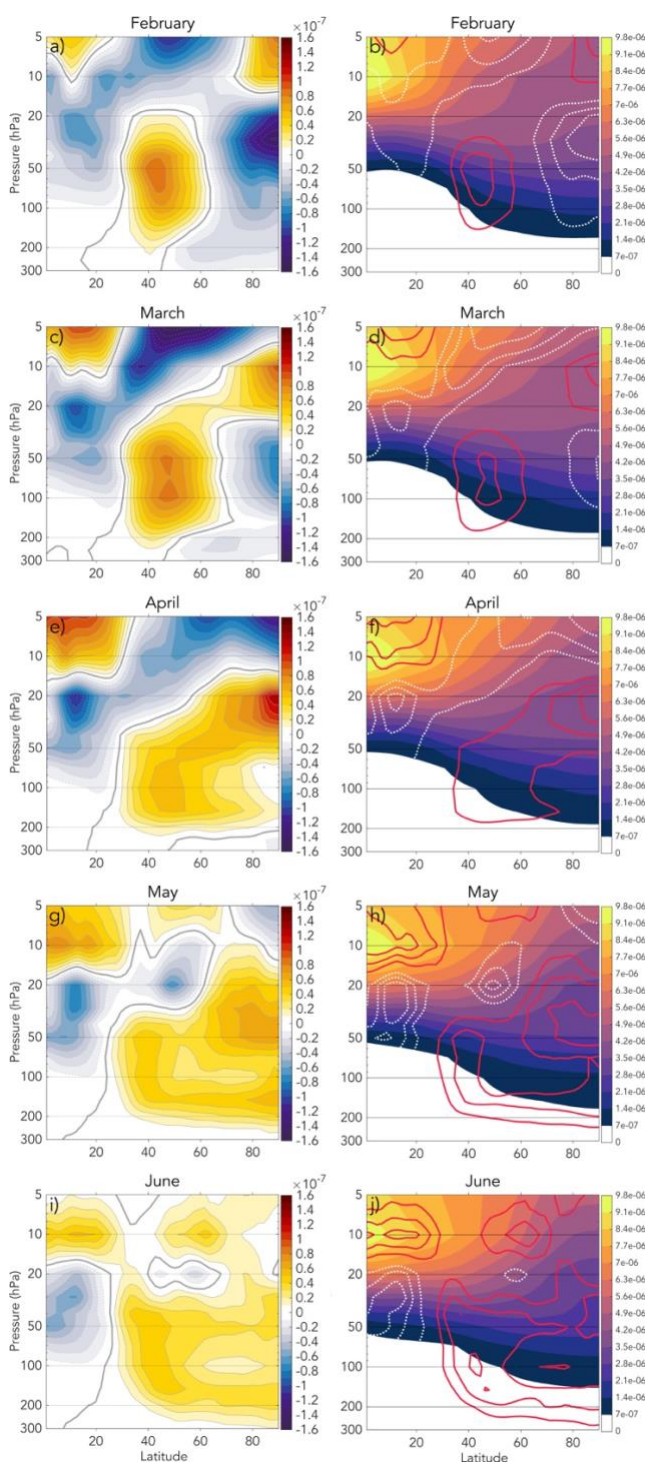

**Figure 3:** February-June El Niño composites of: (left column) O₃S anomalies, and (right column) climatological O₃S (filled contours) with positive (red contours) and negative (white dashed contours) O₃S anomalies from the left columns overlayed. All units are mass mixing ratio.





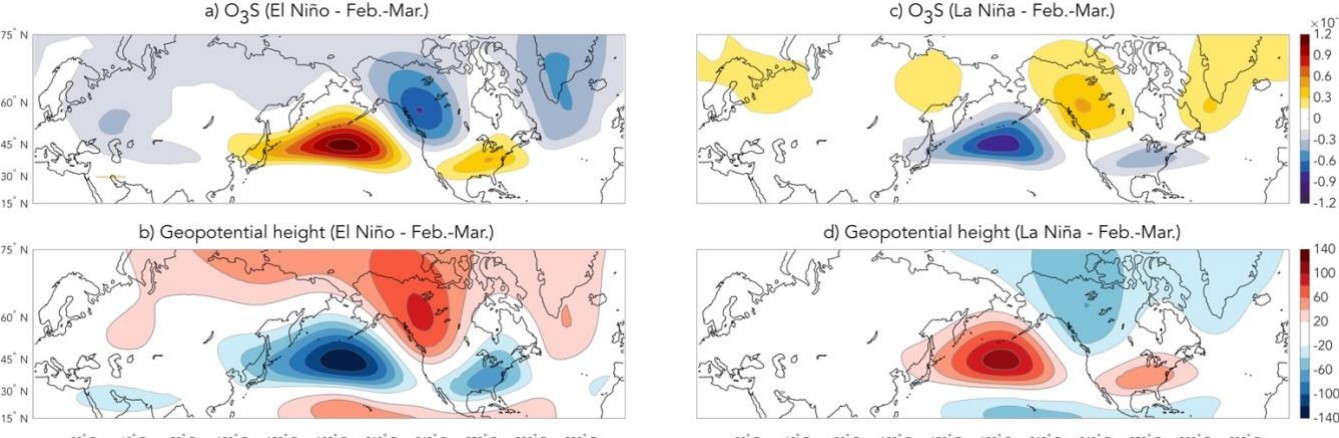

**Figure 4:** February-March (200 hPa pressure level) composites of: O$_3$S (units: mass mixing ratio) for (a) El Niño and (b) La Niña; and geopotential height (units: gpm) for (c) El Niño and (d) La Niña.



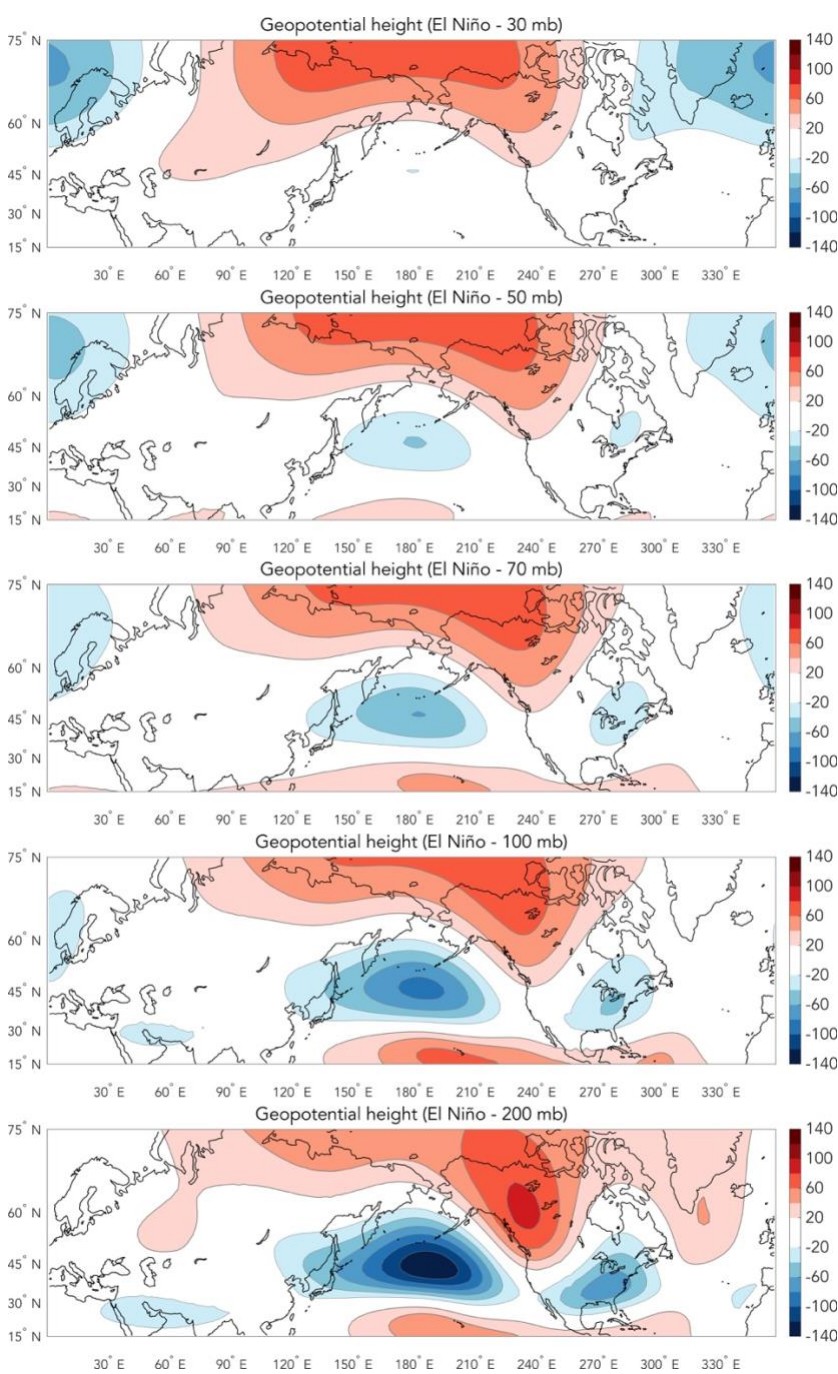

**Figure 5**: February-March El Niño geopotential height (units: gpm) composites between 30-200 mb (top to bottom).


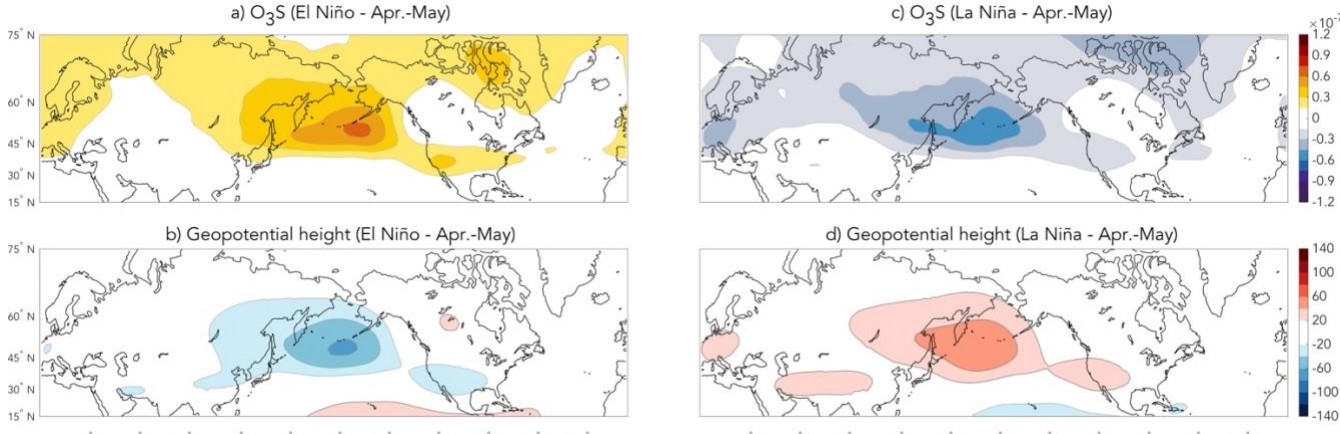

**Figure 6:** April-May (200 hPa pressure level) composites of: O₃S (units: mass mixing ratio) for (a) El Niño and (b) La Niña; and geopotential height (units: gpm) for (c) El Niño and (d) La Niña.

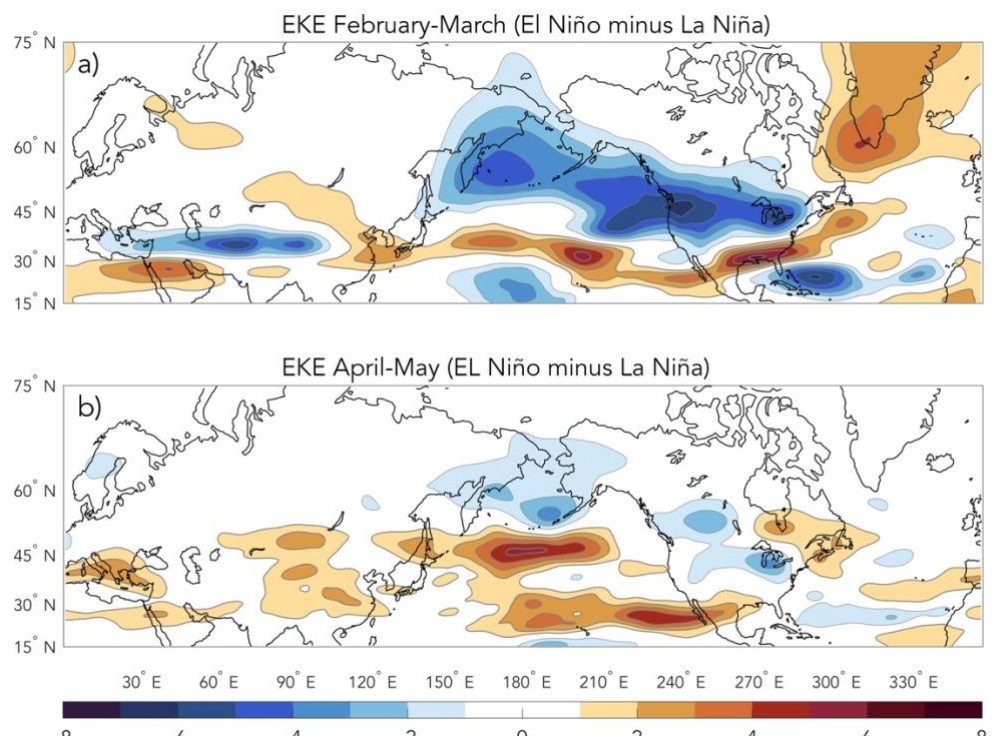

**Figure 7:** 200 hPa pressure level eddy kinetic energy (2 to 10-day filtered) composites for: (a) February-March (El Niño minus La Niña), and (b) April-May (El Niño minus La Niña). Units are: m² s⁻².



**Figure 8:** February-June El Niño composites of: (left column) 800 hPa $O_3S$ anomalies, and (right column) zonal mean anomaly cross-sections of $O_3S$ where the longitudinal boundaries (235°-260° E) of the zonal mean correspond to the box overlaying the left column composites. All units are mass mixing ratio. White areas reflect missing data associated with topography.