# Peer review of "Dynamics of ENSO-driven stratosphere-to-troposphere transport of ozone over North America"

_Atmospheric Chemistry and Physics, 2022_

## Author Comment (AC1)

*Response to Reviewer #1's Comments on:*
"***Dynamics of ENSO-driven stratosphere-to-troposphere transport of ozone over North America***"
by J. Albers et al. (ACP paper # acp-2022-276)

We thank Reviewer #1 for their comments and suggested references, which we have addressed below.

**Reviewer wrote**: *I am not convinced about the implied duality BDC versus ENSO-teleconnection that is presented in the paper (e.g. L135-159, L156-159). The longitudinally-resolved ozone anomalies clearly reveal the ENSO teleconnection pattern associated with the stationary Rossby wave train (PNA). However, this does not imply that the BDC does not play a role in driving ozone anomalies, if the BDC is defined as the combined net zonal mean tracer transport by residual circulation and mixing (as is the case in this paper). In particular, the zonal anomalies combined with circulation anomalies are included in the eddy transport/mixing component of the BDC. In my opinion, the apparent duality BDC/teleconnection duality only results from the different framework (zonal mean/zonally resolved), but does not constitute a different process.*

**Our response**: The reviewer brings up an excellent point, one which the co-authors discussed at some length before submitting the manuscript. We agree that the tracer transport equation (e.g., the TEM formalism as defined in say Sect. 9.4 of Andrews et al. 1987) should in principle form a closed budget for any transport (with the potential exception of unresolved sources not accounted for in WACCM), thus if one simply defines the BDC in terms of the transport equation (i.e., transport from mass entrance at the tropical tropopause to mass exit back into the troposphere at high latitudes), then the ozone teleconnections are not a separate physical process. That said, we believe that there are some important distinctions between the physical processes involved in what is "typically" thought of as the BDC (the residual circulation and isentropic mixing) versus the ozone teleconnections discussed here. In particular, most review papers on stratospheric transport specifically discuss mid-stratospheric extratropical isentropic mixing and advective transport in terms of EP-flux divergence and wave breaking associated with planetary scale waves #1 and 2 that are (mostly) freely propagating into the interior of the stratosphere (e.g., Fig. 2 of Plumb 2002, https://www.jstage.jst.go.jp/article/jmsj/80/4B/80_4B_793/_article). Of course a different spectrum of waves drives the shallow branch of the BDC in the tropics, and gravity waves drive a large fraction of the mesospheric portion of the deep branch, but those waves are not relevant to this discussion. In contrast, the transport discussed here appears to be associated with waves with wavenumber >2, which are largely barotropic (and largely evanescent) above the tropopause (see Figure 5) and thus do not neatly fit into that paradigm. As a (somewhat puzzling) consequence of the waves being barotropic and largely evanescent, the waves should not be strongly violating nonacceleration conditions, which in turn means that they should not be driving large (permanent) changes in the residual circulation (what Andrews et al. refer to as the 'nontransport theorem'). Indeed, the observation that medium scale waves like the ones we focus on here can cause ozone anomalies that are largely *reversible* has been discussed by previous authors, for example, Salby and Callaghan 1993 and Fusco and Salby 1999. However, just because the lower stratospheric transport anomalies are reversible *in isolation*, does not mean that they are not important to STT of ozone. That is, even if the stratospheric anomalies relax back to zero when the ENSO forcing dissipates, if any irreversible processes mix ozone downwards into the troposphere, then this secondary mixing constitutes an important pathway for ENSO to modulate STT.

While we believe that the distinction made above is important, we don't mean to make an argument about what processes should or should not be included in any definition of what the BDC is; rather we just want to make clear that the ozone teleconnections, which are associated with barotropic (evanescent) waves, appear to drive significant zonally asymmetric ozone anomalies that impact STT, with the sign of the STT determined by the locations of the high and low pressure centers of the geopotential anomalies. To clarify this position, we have added text in several places, including:

- Line 18 of the abstract – we have added the word 'other' prior to the words "…ENSO-driven changes in the Brewer-Dobson circulation…", which helps to clarify that the ozone teleconnection-related transport may also be considered a BDC process.
- Line 60-61 – we have changed the wording here to infer that the residual circulation and isentropic mixing may not be the only physical processes that constitute the BDC
- Lines 203-211 – we added a series of sentences here that clarify the role of reversible versus irreversible transport
- Most importantly, on lines 282-319 of the Conclusions, we now greatly expand upon the discussion of reversible versus irreversible transport and point out that if the BDC is broadly defined as any transport from

mass entrance at the tropical tropopause to mass exit back into the troposphere at high latitudes, then the ozone teleconnections are just one aspect of the BDC

**Reviewer wrote:** *151-159: why focus on these high level anomalies when you are interested in STT and there are strong anomalies at lower levels? (also L268)*
**Our response**: Previous studies (including our own, e.g., Albers et al. 2018) focused on the ways that the residual circulation and isentropic mixing might modulate high latitude lower stratospheric ozone, and hence STT. However, we are showing here that it is not these aspects of the BDC that allow ENSO to modulate STT of ozone during the spring. We feel that this is a very important conclusion from our work.

**Reviewer wrote**: *L224-225 "what is important here, is that anomalies in EKE are well-correlated with anomalous STT (Shapiro 1980, Langford 1999)". I believe the two publications cited do not use EKE. Shapiro 1980 deals with turbulent/mixing fluxes and Langford 1999 actually shows the absence of correlation between monthly mean ozone and the eddy variance of meridional wind (their Fig 3d). I suggest to cite Breeden et al. 2021 instead, as they explicitly show the correlation between fold frequency and EKE, which in turn is correlated with STT (their Fig. 7).*
**Our response**: Thank you for noticing that section, indeed our language was a little imprecise. Regions of elevated EKE typically coincide with the locations of jets, which, as you point out, is discussed in Breeden et al. (2021). And since waves are guided along the jet, any process that results in STT will therefore correspond with regions of elevated EKE. The reason for the Shapiro and Langford citations is that they highlight two different types of exchange related to the different jets. In the case of the Langford 1999 citation, he is discussing enhanced STT at the jet exit region of the subtropical jet and the elevated EKE over Baja California; our Fig. 8 is clearly related to the extension of the subtropical jet during El Nino. The enhanced EKE over the North Pacific/northwestern US is related to the northward perturbation of the polar front or midlatitude jet that occurs during La Nina, which should be accompanied by elevated tropopause folds and eddy mixing, hence the Shapiro reference. We have added clarifying text to this section (see lines 244-247) to be more clear about the above chain of reasoning (the new text includes the Breeden et al. 2021 reference as you suggested).

**Reviewer wrote**: *L93-95: Do I interpret this correctly that in your simulations there are continuously varying SST that evolve with the prescribed 2 years cycle (after one cycle ends, the next one starts)? Maybe this could be clarified.)*
**Our response**: Yes, it is a continuously evolving two-year cycle, but the cycle is periodic and goes to zero at the beginning and end of the cycle. We did this so that the runs include the growth and decay stages of ENSO, which we thought would be important, particularly during spring. The two year cycle of SST anomalies is shown in Supplement Figure S1, which should make clear to readers how the SST cycle works. We have added a clarifying sentence on lines 99-100.

**Reviewer wrote**: *96-97: but you said it was constructed averaging all events? This seems to contradict it.*
**Our response**: As stated on line 94, we build our composite from events where ENSO is "…defined as Niño 3.4>1 standard deviation from the March long-term mean…", which means that we are using "all events" that meet this criteria, that is, the strongest (measured by the march standard deviation) March ENSO events. Thus there is no contradiction.

**Reviewer wrote**: *L103: since the simulations have no QBO, are there unrealistic climatological winds in the equatorial stratosphere?*
**Our response**: Because there is no QBO in this version of WACCM, you are correct, the winds in the equatorial stratosphere are unrealistic. But this is typical of climate simulations of this type. Indeed having a realistic QBO in a climate model is still a difficult modeling issue (see for example, the QBO intercomparison project https://doi.org/10.1029/2019JD032362 ). It is a shortcoming of models to not include the QBO, but there is nothing that we can do about that.

**Reviewer wrote**: *104-105: Does this assume that El Niño and La Niña are symmetric (which they are not)? Or is it equivalent to using a climatology with zero SST anomalies (neutral ENSO)?*
**Our response**: In our simulations El Nino and La Nina are defined to be symmetric. This was done to make the simulations simpler and easier to interpret (for example, it avoids trying to interpret how different flavors of ENSO modulate our results). So you are correct when you say that the climatology defined in the way we have constructed it equates to ENSO neutral conditions (really zero SST anomaly because we have so many years in the time slice simulations).

**Reviewer wrote**: - *109: why is longer-timescale variability not relevant near the tropopause?*

**Our response**: Longer timescale variability may be relevant for some processes near the tropopause, but for the 2-10 day filtered EKE, we are intending to construct a proxy that identifies jet location variability (e.g., Breeden et al. 2021). Extending the tropopause level filter out to 120 days would make the physical process we are trying to understand more difficult to isolate.

**Reviewer wrote**: *L149: this is true at levels above 20 hPa or so, but the opposite is seen in the lwoer stratosphere! Again, I think you should focus on the dipole at lower levels.*

**Our response**: As stated in our response to one of your earlier questions, previous studies (including our own, e.g., Albers et al. 2018) focused on the ways that the residual circulation and isentropic mixing might modulate high latitude lower stratospheric ozone, and hence STT. However, we are showing here that it is not these aspects of the BDC that allow ENSO to modulate STT of ozone during the spring. We feel that this is a very important conclusion from our work, thus we will keep this portion of our analysis in the manuscript.

**Reviewer wrote**: *198-199: yes, but the relevant information here is how this seasonal climatological behavior is modulated by ENSO, please include this information with references.*

**Our response**: The focus of our paper is on how much stratospheric ozone is mixed downwards to the middle to lower troposphere, and in particular, how that varies by month. Thus, what is important to our current work is how much ozone is available for transport downward, which Figures 3, 4, and 6 and the corresponding discussions clearly address. The effect of the anomalies shown in Figures 3, 4, and 6 are discussed in the context of Figure 8 where we discuss the ozone that is transported to the troposphere. Thus, we feel that any additional discussion of the breakdown of the polar vortex transport barrier is beyond the scope of this paper. Regarding adding references, we are unaware of any papers that specifically discuss how ENSO modulates the breakdown of the polar vortex mixing barrier. In fact, even the recent ENSO-stratosphere teleconnection review paper by Domeisen et al. 2019 (https://doi.org/10.1029/2018RG000596), which a co-author of this paper is on, does not discuss this. The only paper that we can find that mentions the mixing barrier in the context of ENSO at all is a paper by Benito-Barca et al. (https://doi.org/10.5194/acp-2022-378), which is currently only a discussion paper, and even that paper does not address this topic beyond a single sentence mentioning the barriers existence. We would be happy to add any references that the reviewer knows of.

**Reviewer wrote**: *L232-234: Writing should be more careful here: one needs to compare the ozone anomalies in Fig 4a with EKE anomalies in Fig 7a (both at 200 hPa). Then the area of enhanced ozone in the North Atlantic and that of reduced EKE overlap. In Fig 8a (800 hPa) the area of enhanced ozone is centered at lower latitudes (~30-45N), and this corresponds to enhanced, not reduced, EKE in Fig 7a.*

**Our response**: The mixing that occurs in the North Pacific is related to wave breaking along isentropes (though as pointed out by Shapiro 1980, cross-isentropic processes are critically important as fold development proceeds). Because the isentropes slope downwards from pole to equator (e.g., Figure 2 in Gettelman et al. 2011, https://doi.org/10.1029/2011RG000355), that means that ozone transport associated with PV folds etc. should evolve equatorward and downward, at least initially (see for example, the rightmost columns in Figure 8). Thus, it is expected that ozone anomalies at 200 hPa (like those shown in our Fig. 4) should be somewhat poleward of the lower tropospheric anomalies (like those shown in our Fig. 8). The fact that the EKE anomalies at 200 hPa do not *exactly* line up with the ozone anomalies at 200 hPa (and we would argue that they do line up rather well, if not perfectly), does not negate that interpretation.

That said, it is true that there is enhanced 200 hPa EKE just north of Hawaii (at roughly 30° N). However, at 30° N, the isentropes near to the 200 hPa surface (roughly the 350 theta surface) are relatively flat (i.e., the isentropes do not curve downwards to the surface), which means that most mixing is vertically shallow and will lead to ozone anomalies in the subtropical upper troposphere as discussed by Waugh and Polvani (GRL 2000, https://doi.org/10.1029/2000GL012250). Thus, we feel that there is not any inconsistency with what we write in this section. Nevertheless, we have added several sections of text pointing out: (1) why one would expect the lower tropospheric anomalies should be somewhat equatorward of the stratospheric anomalies (lines 255-260), (2) that we cannot rule out the contribution from transverse circulations to the lower tropospheric anomalies (lines 261-263), and (3) isentropic mixing at 30N is unlikely to be contributing to the surface ozone anomalies over the central Pacific (lines 263-267). Discussing the North Atlantic is not the focus of this manuscript, and are small relative to those over the Pacific basin and North America, so we prefer not to discuss it because we feel that will be a distraction.

**Reviewer wrote**: *L232-233: "which in isolation should correspond to a reduction in tropopause fold frequency" again you could cite Breeden here.*
**Our response**: Citation added.

**Reviewer wrote**: *Fig. 1: The figure caption should include the word "anomalies".*
**Our response**: Fixed (also fixed in other figure captions).

**Reviewer wrote**: *Fig. 5: Is this figure really needed?.*
**Our response**: Yes, this figure depicts the barotropic structure of the waves, which is a critical piece of information.

**Reviewer wrote**: *110: not that slightly.*
**Our response**: Word removed.

**Reviewer wrote**: *L148: EXTENDING from the North Atlantic... (otherwise it is unclear).*
**Our response**: Done.

**Reviewer wrote**: *L174: responsible for THE high latitude....*
**Our response**: Fixed.

**Reviewer wrote**: *L214: typo: -0.5oC (minus sign missing)*
**Our response**: Fixed.

**Reviewer wrote**: *L236: Baja CALIFORNIA*
**Our response**: Fixed there and elsewhere.

**Reviewer wrote**: *L267: DOMINANT*
**Our response**: Text rewritten.

*Response to Reviewer #2's Comments on:*
"***Dynamics of ENSO-driven stratosphere-to-troposphere transport of ozone over North America***"
by J. Albers et al. (ACP paper # acp-2022-276)

We thank Reviewer #2 for their comments and suggested references.

**Reviewer wrote:** *119 "anomalies": Figure S2 only shows climatological values and not anomalies (if I read the figure S2 caption correctly).*
**Our response**: Thank you for carefully reading and noticing that. Fixed.

**Reviewer wrote**: *149 - 151 You may want to mention the tracer tendency equation and cite it (if this is what you mean here by referring to the tracer gradient): O3/dt is proportional to the ~ -v\*(partial dO3/partial dy) + mixing terms ...*
**Our response**: We have added an explanatory sentence and reference to the sections in Andrews et al. (1987) where readers can find more detailed information about the relationship between eddy flux divergences, the meridional gradient of ozone (or any tracer), and ozone/tracer transport.

**Reviewer wrote:** *158 "in the lowermost stratosphere" Please provide pressure levels range (70 -100 hPa ?) here for reference as you did above to be consistent?*
**Our response:** You are right, stating 70-100 hPa is much clearer to readers than just writing 'lower stratosphere'. Fixed.

**Reviewer wrote**: *162 "anomalies": add at which levels*
**Our response:** To clarify we have added (100-300 hPa) after stating 'lowermost stratosphere' to make it clear what region we are referring to.

**Reviewer wrote**: *160 – 173: The idea of ENSO teleconnection needs to be explained better. In particular, what is the role of vertically deep teleconnections in the STT variability? I think what you mean here is that this ENSO teleconnection modifies the amount of available ozone for STT, which strongly depends on the time of the year.*
**Our response**: We have added the following sentence to the end of the paragraph that you reference:

*"The primary role of the ozone teleconnections is to modulate the availability of ozone in the lowermost stratosphere that is available for subsequent transport into the middle to lower troposphere via tropopause folds, potential vorticity streamers and cutoffs (Reed and Danielson, 1958; Hoerling et al., 1993; Langford and Reid, 1998; Shapiro, 1980; Sprenger et al., 2007; Škerlak et al., 2015), and transverse circulations in jet exit regions (Lang- ford et al., 1998; Langford, 1999)."*

The following paragraph carefully describes (with a number of references) the dynamics of stratospheric transport associated with ozone teleconnections.

**Reviewer wrote**: *Figure 2: Please add a few latitude circles to make it easier to locate information from the text.*
**Our response:** Done.

**Reviewer wrote:** *All figures: increase the font size for all figure labels and axis*
**Our response:** Done except for Figs. 1, 3, and 7, where we feel the text size is appropriate.

**Reviewer wrote:** *174: "30-200hPa": change to 20-200hPa to be consistent with numbers in line: 169*
**Our response:** Done.